# INVERSE LEARNING WITH EXTREMELY SPARSE FEEDBACK FOR RECOMMENDATION

## ABSTRACT

Negative sampling is widely used in modern recommender systems, where negative data is randomly sampled from the whole item pool. However, such a strategy often introduces false-negative noises. Existing approaches about de-noising recommendation mainly focus on positive instances while ignoring the noise in the large amount of sampled negative feedback. In this paper, we propose a meta learning method to annotate the unlabeled data from loss and gradient perspectives, which considers the noises on both positive and negative instances. Specifically, we first propose *inverse dual loss* (IDL) to boost the true label learning and prevent the false label learning, based on the loss of unlabeled data towards true and false labels during the training process. To achieve more robust sampling on hard instances, we further propose *inverse gradient* (IG) to explore the correct updating gradient and adjust the updating based on meta learning. We conduct extensive experiments on a benchmark and an industrially collected dataset where our proposed method can significantly improve AUC by 9.25% against state-of-the-art methods. Further analysis verifies the proposed inverse learning is model-agnostic and can well annotate the labels combined with different recommendation backbones. The source code along with the best hyper-parameter settings is available at this link: `https://anonymous.4open.science/r/InverseLearning-4F4F`.

## 1 INTRODUCTION

As one of the most successful machine learning applications in industry, recommender systems are essential to promote user experience and improve user engagement (Ricci et al., 2011; Xue et al., 2017; Liu et al., 2010b), which are widely adopted in online services such as E-commerce and Mirco-video platforms. Aiming to capture users' preference towards items based on their historical behaviors, existing recommenders generally focus on explicit or implicit feedback. Specifically, explicit feedback (Liang et al., 2021) refers to rating data that represents the user preference explicitly. However, collecting sufficient explicit data for recommendations is difficult because it requires users to actively provide ratings (Jannach et al., 2018). In contrast, implicit feedback, such as user clicks, purchases, and views (Liang et al., 2016), is much richer (Liu et al., 2010b) and frequently used in modern recommender systems (Chen et al., 2020). Particularly, in feed recommendation for online platforms such as Micro-video, users are passive to receive the recommended items without any active clicking or rating action. That is to say, we have a large number of unlabeled feedback, with extremely sparse labeled data, which becomes a key challenge for recommendation.

To tackle the unlabeled feedback, most works (He et al., 2017; Chen et al., 2019) randomly sample unlabeled data and treat it as negative feedback, resulting in unavoidable noise. Specifically, the collected user click data is often treated as positive feedback, and the unclicked data is sampled as negative feedback (He et al., 2017; Chen et al., 2019). However, there may be some positive unlabeled data sampled by the negative sampling strategy, which means that these instances will be false-negative. There are also some works about hard negative sampling which will decrease the false-positive but increase the false-negative instances (Zhang et al., 2013; Ding et al., 2019; 2020). These hard negative methods tend to perform poorly when tested on both true positive and negative data instead of true positive but sampled negative data (as shown in Appendix 5). A recent work DenoisingRec (Wang et al., 2021) denoises positive feedback by truncating or reweighing the loss of false-positive instances without further consideration of the noisy negative feedback. In general,

existing works either focus only on the positive or the negative perspectives. To resolve the data-noise problem from both positive and negative perspectives at the same time, we propose a novel *Inverse Learning* approach that automatically labels the unlabeled data via inverse dual loss and inverse gradient, assuming there are both positive and negative feedbacks in the unlabeled data. Firstly, based on the property that the loss of a false positive/negative instance will be larger than that of a true positive/negative instance (Wang et al., 2021), we assign both positive and negative labels to the unlabeled instance with different weights calculated by inverse dual loss. Specifically, we assign larger (smaller) weight to the label with smaller (larger) loss value. In this way, we take full advantage of true positive/negative instances, and eliminate the noise of false positive/negative instances. To achieve more robust sampling on hard instances, we further propose an inverse gradient method. Here we build a meta-learning process (Finn et al., 2017; Li et al., 2017) and split the training data into training-train and training-test data. We first exploit training-train data to pre-train the model. Then we further use training-test data to validate the correctness of classification on the sampled instance. Specifically, we calculate the gradient for the inverse dual loss of sampled instances as well as the additive inverse of the gradient. The model is optimized by either the direct gradient or the additive inverse of gradient, determined by the split training-test data. Experimental results illustrate that inverse gradient can truly improve the inverse dual loss. In summary, the main contributions of this paper are as follows:

- To the best of our knowledge, we are the first to sample both positive and negative data in recommendation, which is far more challenging while existing works only sample negative data.

- We propose inverse dual loss to learn the label for sampled instances and further exploit inverse gradient to adjust the false label for hard instances.

- We experiment on two real-world datasets, verifying the superiority of our method compared with state-of-the-art approaches. Further studies sustain the effectiveness of our proposed method in label annotation and gradient descent.

## 2 PROBLEM FORMULATION AND OUR APPROACH

In this section, we will firstly formulate the problem and perform in-depth analysis of existing solutions and their limitations. Then we will propose inverse dual loss to address the limitations of existing works for easy samples. Finally, we further propose inverse gradient to address the limitation of inverse dual loss and make it capable for not only easy samples but also hard samples.

### 2.1 PROBLEM DEFINITION

The recommendation task aims to model relevance score $\hat{y}_{ui}^{\boldsymbol{\theta}} = f(u, i|\boldsymbol{\theta})$ of user $u$ towards item $i$ under parameters $\boldsymbol{\theta}$. The *LogLoss* function (Zhou et al., 2018; 2019) function to learn ideal parameters $\boldsymbol{\theta}^*$ is as:

$$\mathcal{L}_{\mathcal{D}^*}(\boldsymbol{\theta}) = \frac{1}{|\mathcal{D}^*|} \sum_{\left(u, i, y_{ui}^*\right) \in \mathcal{D}^*} \ell\left(\hat{y}_{ui}^{\boldsymbol{\theta}}, y_{ui}^*\right), \tag{1}$$

where $\ell\left(\hat{y}_{ui}^{\boldsymbol{\theta}}, y_{ui}^*\right) = -\left(y_{ui}^* \log\left(\hat{y}_{ui}^{\boldsymbol{\theta}}\right) + (1 - y_{ui}^*) \log\left(1 - \hat{y}_{ui}^{\boldsymbol{\theta}}\right)\right)$, $y_{ui}^* \in \{0, 1\}$ is the feedback of user $u$ towards item $i$. $\mathcal{D}^* = \{(u, i, y_{ui}^*) \mid u \in \mathcal{U}, i \in \mathcal{I}\}$ is the reliable interaction data between all user-item pairs. Indeed, due to the limited collected feedback, the model training is truly formalized as: $\bar{\boldsymbol{\theta}} = \arg\min_{\boldsymbol{\theta}} \mathcal{L}_{\mathcal{D}^l}(\boldsymbol{\theta}) + \mathcal{L}_{\mathcal{D}^u}(\boldsymbol{\theta})$, where $\mathcal{D}^l \sim \mathcal{D}^*$ is the collected labeled data, and $\mathcal{D}^u = \{(u, i, \bar{y}_{ui}) \mid u \in \mathcal{U}, i \in \mathcal{I}\}$ is the sampled unlabeled data where $\bar{y}_{ui} = 0$ is often assumed in existing recommenders for negative sampling. However, such a strategy will inevitably introduce noise because there are some positive unlabeled instances in the sampled data. As a consequence, a model (i.e., $\bar{\boldsymbol{\theta}}$) trained with noisy data tends to exhibit suboptimal performance. Thus, our goal is to construct a denoising recommender approximating to the ideal recommender $\boldsymbol{\theta}^*$ as:

$$\boldsymbol{\theta}^* = \arg\min_{\boldsymbol{\theta}} \mathcal{L}_{\mathcal{D}^l}(\boldsymbol{\theta}) + \mathcal{L}_{\mathcal{D}^u}^{\text{denoise}}(\boldsymbol{\theta}), \tag{2}$$

where $\mathcal{L}_{\mathcal{D}^u}^{\text{denoise}}(\boldsymbol{\theta})$ indicates the loss on unlabeled data with all samples annotated correctly, *i.e.* denoising sampling.

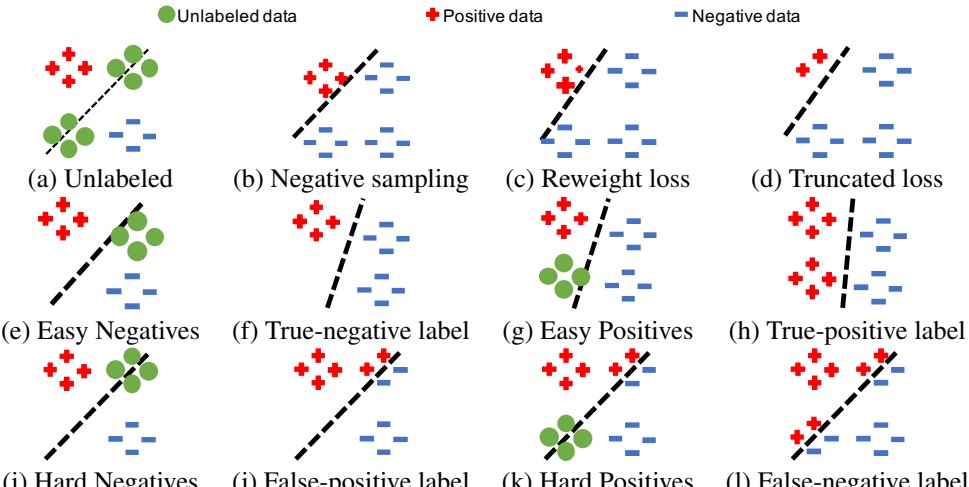

Figure 1: Illustrations of existing solutions, and our inverse dual loss's effectiveness and limitation. **(a)-(d) are the illustrations of existing solutions**: (a) illustrates there are lot of unlabeled data; (b) illustrates the traditional negative sampling approach; (c) illustrates the reweighted loss of DenoisingRec; (d) illustrates the truncated loss of DenoisingRec adapted on the false-negative instance. **(e)-(f) are the illustrations of our inverse dual loss's effectiveness with easy sampling**: (e) illustrates the easy negative instances are sampled; (f) illustrates labeling the sampled instances as true negative; (g) illustrates the easy positive instances are sampled; (h) illustrates labeling the sampled instances as true positive and approximate to **ground-truth**. **(i)-(l) are the illustrations of our inverse dual loss's limitation with hard sampling**: (i) illustrates the hard negative instances are sampled; (j) illustrates labeling part of the sampled instances as false positive; (k) illustrates the hard positive instances are sampled; (l) illustrates labeling part of the sampled instances as false negative.

## 2.2 INVERSE DUAL LOSS

In this section, we first analyze the characteristics of existing solutions on the sampled unlabeled data. Then we introduce the proposed inverse dual loss solution to denoise sampled data.

### 2.2.1 ANALYSIS OF EXISTING APPROACH

We first explain the data sparsity problem in recommender systems from the perspective of classification boundary, based on which we will introduce the existing solutions. As shown in Figure 1 (a), in recommender systems, labeled data tends to be extremely sparse compared with a large number of unlabeled data. A recommendation model is prone to overfitting if it is only trained based on the sparse labeled data, compared with the ground-truth in Figure 1 (h).

In practice, existing recommenders often sample from unlabeled data and treat all the sampled data as negative feedback. Such an approach introduces false negative, which fails to retrieve items that users may be interested in, as shown by the classification boundary in Figure 1 (b). That is, there exists noise in the sampled negative data. However, existing denoising approaches mainly focus on the noise in positive samples (false positive). For example, DenoisingRec (Wang et al., 2021) attempts to achieve denoising for false positive instances as:

$$\boldsymbol{\theta}^* = \arg\min_{\boldsymbol{\theta}} \mathcal{L}^{\text{denoise}}_{\mathcal{D}^l \cup \mathcal{D}^{noise}}(\boldsymbol{\theta}) + \mathcal{L}_{\mathcal{D}^u}(\boldsymbol{\theta}) \tag{3}$$

where $\mathcal{D}^{noise} = \{(u, i, 1) \mid u \in \mathcal{U}, i \in \mathcal{I}, y^*_{ui} = 0\}$ is the noisy false positive data they introduce in experiments. For example, R-CE (Reweight Cross-Entropy) of DenoisingRec assigns lower weight on false-positive instances with large loss (Figure 1 (c)), and T-CE (Truncated Cross-Entropy) of DenoisingRec discards those false positive instances with large loss (Figure 1 (d)). Though achieving denoising for false-positive instances, they ignore false-negative instances and fail to address the noise brought by the negative sampling.

In fact, the collected labeled data is much cleaner than sampled unlabeled data, and the number of false-positive instances is limited in real-world recommender systems. On the contrary, the noise brought by negative sampling is far more harmful. In other words, the noise level of positive unlabeled data incorrectly sampled as negative feedback is much higher than that of negative samples wrongly regarded as positive feedback.

To sum up, existing solutions either introduce noise or perform incomplete denoising, which motivates us to propose a denoising solution for unlabeled data.

### 2.2.2 LABELING WITH INVERSE DUAL LOSS

As an existing attempt in DenoisingRec, we have discovered that the false positive instances are with a greater loss. It is also an apparent phenomenon in machine learning. For example, if we have a positive instance and a well-trained model, the loss of classifying it as negative will be greater than that of classifying it as positive. Otherwise, if we have a negative instance and a well-trained model, the loss of classifying it as positive will be greater. Hence, we can assume the sampled unlabeled instances are both possibly positive and negative and then exploit this inherent characteristic to automatically weigh more on the true positive or negative instances while weighing less on the false ones.

**Definition 1** *(Inverse Dual Loss) The inverse dual loss is defined as denoising loss to automatically classify the unlabeled data as:*

$$\mathcal{L}_{\mathcal{D}^u}^{denoise}(\boldsymbol{\theta}) = \mathcal{L}_{\mathcal{D}^u}^{dual}(\boldsymbol{\theta}) = \frac{1}{|\mathcal{D}^u|} \sum_{(u,i,\bar{y}_{ui}) \in \mathcal{D}^u} w^1 \ell\left(\hat{y}_{ui}^{\boldsymbol{\theta}}, 1\right) + w^0 \ell\left(\hat{y}_{ui}^{\boldsymbol{\theta}}, 0\right) \quad (4)$$

*where $w^1 = \frac{|\ell(\hat{y}_{ui}^{\boldsymbol{\theta}}, 0)|}{z_w |\ell(\hat{y}_{ui}^{\boldsymbol{\theta}}, 1)|}, w^0 = \frac{|\ell(\hat{y}_{ui}^{\boldsymbol{\theta}}, 1)|}{z_w |\ell(\hat{y}_{ui}^{\boldsymbol{\theta}}, 0)|}$ are the weights for positive loss and negative loss, respectively. $z_w = \frac{|\ell(\hat{y}_{ui}^{\boldsymbol{\theta}}, 0)|}{|\ell(\hat{y}_{ui}^{\boldsymbol{\theta}}, 1)|} + \frac{|\ell(\hat{y}_{ui}^{\boldsymbol{\theta}}, 1)|}{|\ell(\hat{y}_{ui}^{\boldsymbol{\theta}}, 0)|}$ is the normalization parameter. $\mathcal{D}^u$ is the sampled unlabeled data and $\boldsymbol{\theta}$ is the model parameters to be learned.*

The pros of inverse dual loss are as shown in (e)-(h) of Figure 1: (e) when the easy negative instances are sampled, the loss of classifying them as positive will be greater than that of classifying them as negative, and thus inverse dual loss will assign more weights on the negative loss; (f) gradually assigning more and more weights on the negative loss, the negative unlabeled instances will eventually be classified as negative; (g) likewise, when the easy positive instances are sampled, the inverse dual loss will assign more weights on the positive loss; (h) the positive unlabeled instances will eventually be classified as positive and approximate to the ground-truth.

### 2.2.3 LIMITATION OF INVERSE DUAL LOSS

When sampling the easy positive or negative instances, our inverse dual loss can boost the learning via correctly labeling the sampled instances. However, it may be an obstacle when there are some hard positive or negative instances. As shown in (i)-(l) of Figure 1, given some hard positive or negative instances, the classification boundary will be prevented from the ground-truth: (i) the hard negative instances are sampled; (j) half of the negative unlabeled instances will be classified as positive and become noise; (k) the hard positive instances are sampled; (l) half of the positive unlabeled instances will be classified as negative and become noise. That is to say, our inverse dual loss relies heavily on the current training classification boundary and the difficulty of sampled data, requiring us to further improve its robustness.

### 2.3 INVERSE GRADIENT

To improve the robustness of inverse dual loss on the hard sampled data towards the current training model, in this section, we further propose inverse gradient to adjust the gradient of hard sampled data, inspired by the meta-learning framework (Finn et al., 2017; Li et al., 2017). Then we analyze the convergence of the proposed inverse gradient.

### 2.3.1 LEARNING TO LABEL WITH INVERSE GRADIENT

In this part, we introduce our solution for tackling the hard sampled data of inverse dual loss.

**Definition 2** *(Inverse Gradient) We define the gradient and additive inverse of gradient calculated by equation 4 w.r.t.* $\nabla \mathcal{L}_{\mathcal{D}^u}^{dual}(\boldsymbol{\theta})$ *and* $-\nabla \mathcal{L}_{\mathcal{D}^u}^{dual}(\boldsymbol{\theta})$ *as direct gradient and inverse gradient, respectively, of the loss for unlabeled data* $\mathcal{D}^u$.

**Theorem 1** *Given learning rate* $\alpha \in \mathbb{R}, \alpha \neq 0$, *assume the temporal model parameters updated by the direct gradient and inverse gradient, respectively, are as* $\boldsymbol{\theta}^d = \boldsymbol{\theta} - \alpha \circ \nabla \mathcal{L}_{\mathcal{D}^u}^{dual}(\boldsymbol{\theta})$ *and* $\boldsymbol{\theta}^i = \boldsymbol{\theta} + \alpha \circ \nabla \mathcal{L}_{\mathcal{D}^u}^{dual}(\boldsymbol{\theta})$. *Then, the relationship between the loss of them and the model with parameter* $\boldsymbol{\theta}$ *on data* $\mathcal{D}^l$ *will be either* $\mathcal{L}_{\mathcal{D}^l}(\boldsymbol{\theta}^d) > \mathcal{L}_{\mathcal{D}^l}(\boldsymbol{\theta}) > \mathcal{L}_{\mathcal{D}^l}(\boldsymbol{\theta}^i)$ *or* $\mathcal{L}_{\mathcal{D}^l}(\boldsymbol{\theta}^i) > \mathcal{L}_{\mathcal{D}^l}(\boldsymbol{\theta}) > \mathcal{L}_{\mathcal{D}^l}(\boldsymbol{\theta}^d)$.

The proof of Theorem 1 can be found in Appendix A.1. Based on this theorem, we can have the following gradient updating strategies. Generally, we will first split the training data into training-train data and training-test data, where we pre-train the model on the training-train data. Then we will calculate the direct gradient of inverse dual loss on the sampled unlabeled data, which can further results in the following three cases:

- When the sampled data is easy, we can exploit the direct gradient to update the model, and it will gain a smaller test loss on the training-test data, as shown in Figure 2 (a);
- When the sampled data is hard, exploiting the inverse gradient to update the model will gain a smaller test loss on the training-test data, and thus we exploit inverse gradient here as shown in Figure 2 (b);
- When the model is approximately optimal, either direct gradient or inverse gradient will prevent it from ground-truth, and we discard this batch of unlabeled data as shown in Figure 2 (c);

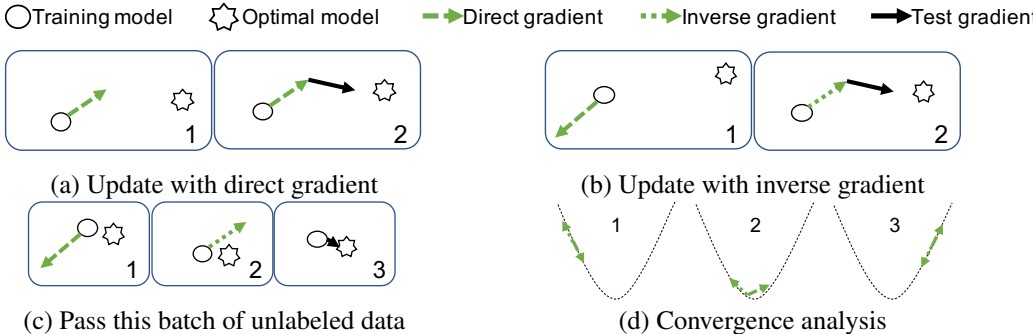

Figure 2: Illustrations of inverse gradient adaption.

Specifically, we present the procedure of exploring these three cases as Algorithm 1 in the Appendix A.2. The algorithm first pre-trains the model using the split training-train data. Then the model will update with direct gradient or inverse gradient or even do not update, determined by the validation on the split training-test data.

### 2.3.2 CONVERGENCE ANALYSIS

As shown in Figure 2 (d), the first case illustrates when the unlabeled data is ideally sampled, the updating with direct gradient will lead to a smaller loss and better convergence on the training-test data, while the third case with poorly sampled data supposes to update with inverse gradient. However, when the model approximates convergence on the training-test data, updating with either direct gradient or inverse gradient may be poorer than no updating as shown in the stage 2 of Figure 2 (d).

To avoid the gradient ascent problem for the second case, we can set the learning rate for the inverse dual loss to be smaller than the test loss, i.e., $\alpha < \gamma$ in Algorithm 1. In this way, the scale of updating by the gradient for inverse dual loss will be within the scale of updating by the gradient for

test loss. That is to say, the gradient ascent problem is less likely to occur on the inverse dual loss for unlabeled data than the loss for labeled data.

## 3 EXPERIMENTS

In this section, we perform experiments on two real-world datasets, targeting three research questions (RQ): (1) How does the proposed method perform compared with the state-of-the-art denoising recommenders? (2) How does our proposed inverse dual loss identify the unlabeled data? (3) What is the effect of the inverse gradient on convergence?

**Datasets** To practice and verify the effectiveness of our proposed method, we conduct experiments on an industrial Micro-Video dataset and a public benchmark ML1M dataset which is widely used in existing work for recommender systems (Lian et al., 2018; Cheng et al., 2020). Micro-Video is the extremely sparse dataset where users are passive to receive the feed videos and have rare active feedback. We introduce the details of these datasets, including the pre-processing steps, in Appendix A.3.

**Baselines and Backbones** To demonstrate the effectiveness of our proposed inverse learning on unlabeled data, we compared the performance of recommenders trained by our inverse gradient (IG) with recommenders trained by inverse dual loss (IDL) and normal training by standard loss or negative sampling (NS) (Rendle et al., 2012; He et al., 2017; Chen et al., 2019). Besides, we also compare our inverse learning method with the state-of-the-art methods for denoising recommender systems. Specifically, we also compare two adaptive denoising training strategies, T-CE and R-CE, of DenoisingRec (Wang et al., 2021).

- Traditional strategies: standard loss without sampling and with negative sampling (NS) (Rendle et al., 2012; He et al., 2017; Chen et al., 2019).
- Denoising strategies: T-CE (Wang et al., 2021) of DenoisingRec that truncates the loss for false-positive instances and R-CE (Wang et al., 2021) of DenoisingRec weighs less on the loss for false-positive instances; Our inverse dual loss that weighs more on the label with smaller loss.

Following DenoisingRec (Wang et al., 2021), we select GMF and NeuMF (He et al., 2017) as backbones, which are neural Collaborative Filtering models.

- GMF (He et al., 2017): A variant of matrix factorization with the element-wise product and a linear neural layer as the interaction function instead of the inner product.
- NeuMF (He et al., 2017): A combination of GMF and Multi-Layer Perceptron (MLP).

**Evaluation Metrics** We adopt widely-used AUC and GAUC as accuracy metrics (Gunawardana & Shani, 2015). Besides, two widely-used ranking metrics (Chang et al., 2021), MRR and NDCG@10, are also adopted for evaluation.

**Hyper-parameter Settings** For the two denoising strategies (Wang et al., 2021), we followed their default settings and verified the effectiveness of our methods under the same conditions. The embedding size and batch size of all models are set as 32 and 1,024, respectively. Besides, we adopt Adam (Kingma & Ba, 2015) to optimize all the model parameters with the learning rate $\gamma$ initialized as 0.0001 and 0.00001 for labeled data on ML1M and Micro-Video datasets, respectively, while the learning rate for sampled data is set as $\alpha = 0.1\gamma$. As for the inverse gradient, we split 90% of the training data as training-train data, and the left is used as training-test data. The sampling rate is set as 1. The provided code has included the best hyper-parameters.

### 3.1 OVERALL PERFORMANCE

The performance comparison is shown in Table 1, from which we have the following observations.

- **Our inverse gradient performs best.** Our inverse gradient (IG) method achieves the best performance compared with four baselines and our inverse dual loss (IDL) for three metrics. Specifically, our IG improves the backbone sharply, which shows the ability of our proposed method to well classify the unlabeled data and achieve effective data augmentation to resolve the data sparsity problem of existing recommenders. Besides, IG outperforms the existing negative sampling

Table 1: Performance comparisons with GMF and NeuMF backbones on two datasets. Bold and underline refer to the best and second best result, respectively. Here IG includes the IDL method.

| Model | Method | ML1M | | | Micro-Video | | |
|---|---|---|---|---|---|---|---|
| | | NDCG | AUC | GAUC | NDCG | AUC | GAUC |
| GMF | None | 0.9285 | 0.7671 | 0.6919 | 0.7365 | 0.8024 | 0.7558 |
| | NS | 0.9400 | 0.7639 | 0.7200 | 0.7049 | 0.7802 | 0.7247 |
| | T-CE | 0.9349 | 0.7612 | 0.7163 | 0.6994 | 0.6486 | 0.6956 |
| | R-CE | 0.9391 | 0.7632 | 0.7192 | 0.7069 | 0.7820 | 0.7260 |
| | IDL | 0.8996 | 0.7304 | 0.6400 | 0.7272 | 0.7858 | 0.7335 |
| | IG | **0.9521** | **0.8318** | **0.7642** | **0.7773** | **0.8033** | **0.7593** |
| NeuMF | None | 0.9214 | 0.7524 | 0.6856 | 0.6649 | 0.7504 | 0.6916 |
| | NS | 0.9298 | 0.7495 | 0.7088 | 0.6350 | 0.7191 | 0.6689 |
| | T-CE | 0.9351 | 0.7587 | 0.7158 | 0.6725 | 0.7469 | 0.6911 |
| | R-CE | 0.9349 | 0.7521 | 0.7117 | 0.6288 | 0.7038 | 0.6568 |
| | IDL | 0.9212 | 0.7962 | 0.6984 | 0.6466 | 0.7174 | 0.6647 |
| | IG | **0.9449** | **0.8253** | **0.7569** | **0.7809** | **0.8198** | **0.7689** |

(NS) method, which means there is truly a large number of positive unlabeled data, and directly treating them all as negative feedback will confuse the model. Finally, IG also outperforms existing state-of-art denoising methods, T-CE and R-CE, showing the importance of tackling the noise from both positive and negative feedback.

- **Inverse gradient can make inverse dual loss more robust.** Inverse dual loss (IDL) only outperforms the NeuMF on ML1M dataset for AUC and GAUC, which shows the inferior robustness of IDL since it depends heavily on the training model and sampled data. Besides, it is even outperformed by GMF on ML1M and NeuMF on Micro-Video, which means the poor sampling of unlabeled data will have a negative impact on the model training. These results show the significance of improving the robustness of IDL and confirm that it is necessary to exploit IG to adjust IDL.

## 3.2 IDENTIFICATION ON UNLABELED DATA

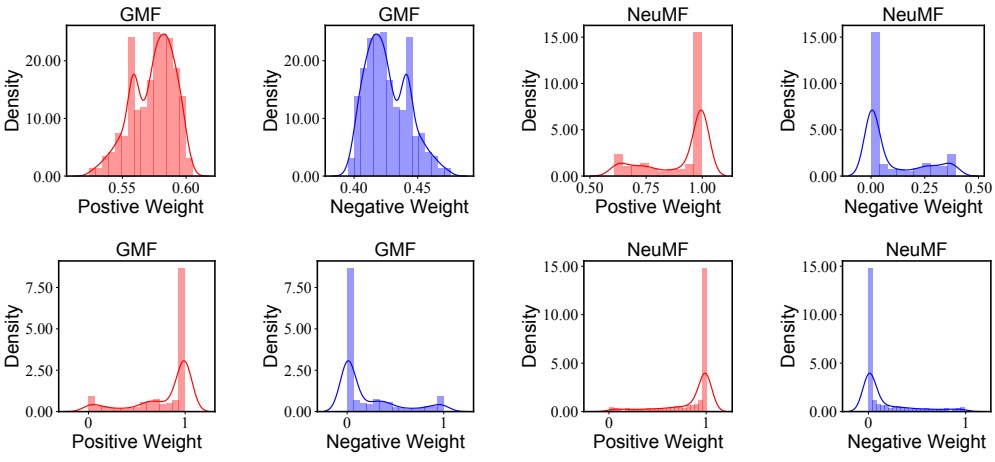

Figure 3: Positive and negative weight distributions for dual loss on ML1M at first (up) and final (bottom) epochs.

We visualize the distribution of weights for positive loss and negative loss on ML1M dataset in Figure 3. In specific, the upper part represents the weights at the first epoch after pre-training on the training-train data, and the bottom part represents the weights at the convergence epoch. From the figure, we can observe that: (1) at the beginning, GMF model fails to well classify the unlabeled data but NeuMF model has well captured the pattern of unlabeled data, which shows the generalization

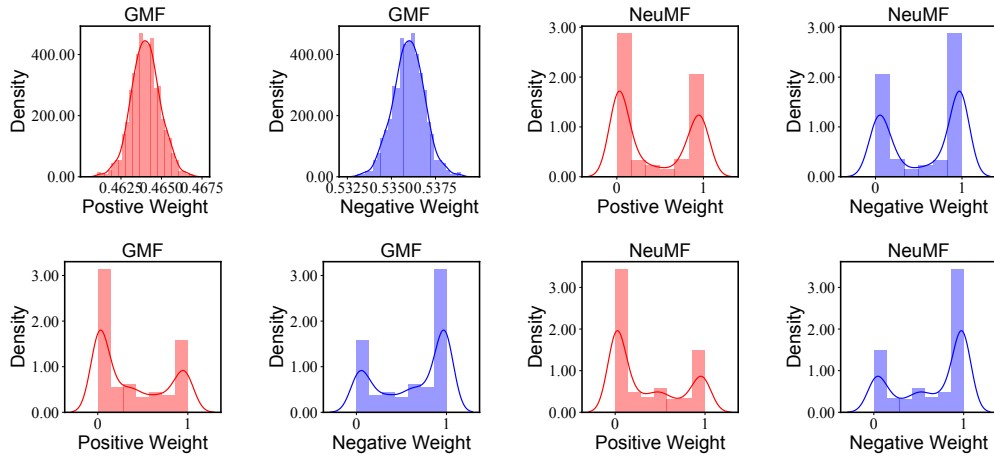

Figure 4: Positive and negative weight distributions for dual loss on Micro-Video at first (up) and final (bottom) epochs.

ability of deep learning model (Cheng et al., 2016); (2) after convergence, both GMF and NeuMF can well classify the unlabeled data and capture the similar pattern where most sampled instances are labeled as positive with more positive weights.

Besides, Figure 4 demonstrates the distribution of weights for positive loss and negative loss on Micro-Video dataset, which shows similar results as that on ML1M dataset despite there being more balanced positive and negative labels on the unlabeled data.

## 3.3 IMPACT OF INVERSE GRADIENT ON CONVERGENCE

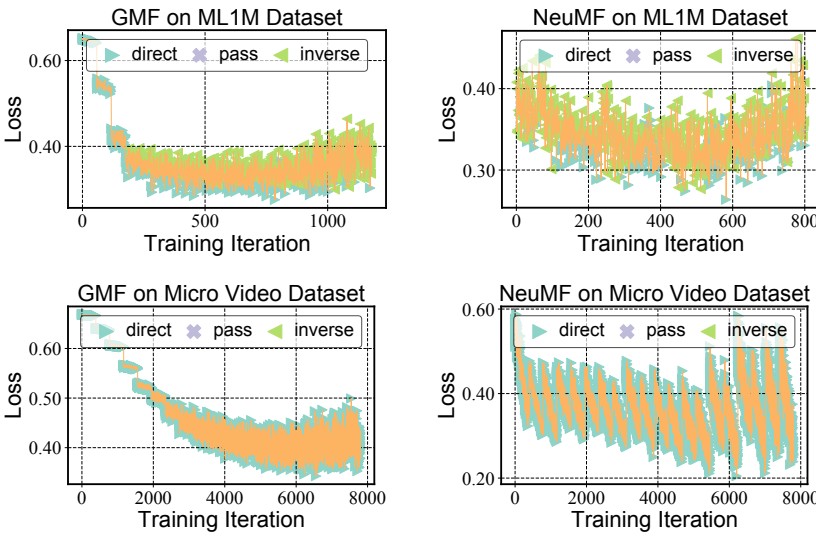

Figure 5: The loss on training-test data with adapted gradient from the dual loss of unlabeled data. Each point of the loss curve is marked by its updated gradient direction.

To investigate the convergence of our proposed inverse gradient, we also plot the loss curve on the training-test data as shown in Figure 5. Based on the results on ML1M dataset, we can discover that: (1) the pre-training on training-train data can promote the model learning effectively at the early stage for both GMF and NeuMF models, while NeuMF will come into a fast convergence in

the first epoch; (2) GMF model is updated with direct gradient, then with a hybrid of direct and inverse gradients, and finally with inverse gradient while in NeuMF model we can discover more hybrid of gradients in the valley of loss curve, this is because the test gradient is more likely to ascent at the valley and thus pushes the gradient for dual loss to adjust its direction. Besides, based on the results on Micro-Video dataset, we can discover that: (1) the models are always updated with direct gradient. This is because the learning rate is relatively low here (as analyzed in the Appendix A.5), leading to almost no gradient ascent problem; (2) the pre-training on the training-train data will conflict with the test data, which means deep learning based model is prone to be overfitting (Cheng et al., 2016) in the sparse data. Most importantly, we can discover that there is no case of passing gradient in the descent procedures, which confirms our analysis at Section 2.3.2 that setting a lower learning rate $\alpha$ can avoid the gradient ascent problem for inverse dual loss.

## 4 RELATED WORK

**Implicit Feedback with Negative Sampling.** Existing recommenders are generally based on implicit feedback data, where the collected data is often treated as positive feedback, and negative sampling (Rendle et al., 2012; He et al., 2017; Chen et al., 2019) is exploited to balance the lack of negative instances. However, the negative sampling strategy will introduce noise because there are some positive unlabeled (Elkan & Noto, 2008; Bekker et al., 2019; Saito et al., 2020) data in the sampled instances. To improve existing implicit feedback recommendation, the identification of negative experiences (Fox et al., 2005; Kim et al., 2014) has grabbed the researchers' attention. However, these methods collect either the various user feedback (e.g., dwell time (Kim et al., 2014) and skip (Wen et al., 2019)) or the item characteristics (Lu et al., 2018), requiring additional feedback and manual labeling, e.g., users are supposed to actively provide their satisfaction. Besides, the evaluation of items relies heavily on manual labeling and professional knowledge (Lu et al., 2018). Thus in practice, these methods are too expensive to implement in real-world recommenders. In addition, hard negative sampling is adopted to improve the negative sampling (Zhang et al., 2013; Ding et al., 2019; 2020). However, with less false positive samples, the hard negative instances also brings more false-negative samples. Our meta-learning method elegantly annotates the unlabeled instances based on the sparse labeled instances.

**Denoising Recommender Systems.** One intuitive approach to reduce noise is to directly include more accurate feedback (Liu et al., 2010a; Yang et al., 2012), such as dwell time (Yi et al., 2014) and skip (Wen et al., 2019)). However, forcefully requiring additional feedback from users may harm the user experiences. To address this problem, DenoisingRec (Wang et al., 2021) achieves denoising recommendation for implicit feedback without any additional data. More specifically, they perform denoising on the false positive instances via truncating or reweighting the samples with larger loss. However, they only consider the positive feedback without further addressing the noise brought by negative sampling. Our work considers the sampled instances as possibly positive and negative, then achieve denoising data augmentation from both positive and negative perspectives.

## 5 CONCLUSIONS AND FUTURE WORK

In this paper, we proposed a novel method that learns the sampled data with both positive and negative feedback. Such exploration not only tackled the unavoidable noise brought by widely used negative sampling but also improved the current denoising recommenders. Specifically, we proposed inverse learning from both loss and gradient perspectives. The first one was the inverse dual loss that assumed the sampled data to be possibly positive or negative. If the positive loss was greater than the negative loss (difficult to label the data as positive), the inverse dual loss would inversely assign more weights to the negative loss and vice versa. Since the inverse dual loss depended heavily on the training model and sampled data, we further proposed inverse gradient which makes inverse dual loss more robust by adjusting the gradient. We designed a meta-learning method with the training data split into training-train data and training-test data. The model was first pre-trained on the training-train data. Then the pre-trained model would explore to update with the gradient or the additive inverse of gradient, or even do not update, determined by the training-test data.

As for future work, we plan to apply our inverse learning with more recommendation models as the backbones to further verify the generalization of our proposed methods.

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

# A APPENDIX

## A.1 PROOF OF THEOREM 1

**Theorem 1** *Given learning rate $\alpha \in \mathbb{R}, \alpha \neq 0$, assume the temporal model parameters updated by the direct gradient and inverse gradient, respectively, are as $\boldsymbol{\theta}^d = \boldsymbol{\theta} - \alpha \circ \nabla \mathcal{L}_{\mathcal{D}^u}^{dual}(\boldsymbol{\theta})$ and $\boldsymbol{\theta}^i = \boldsymbol{\theta} + \alpha \circ \nabla \mathcal{L}_{\mathcal{D}^u}^{dual}(\boldsymbol{\theta})$. Then, the relationship between the loss of them and the model with parameter $\boldsymbol{\theta}$ on data instance $(u, i, y_{ui}^*) \in \mathcal{D}^l$ will be either $\mathcal{L}_{\mathcal{D}^l}(\boldsymbol{\theta}^d) > \mathcal{L}_{\mathcal{D}^l}(\boldsymbol{\theta}) > \mathcal{L}_{\mathcal{D}^l}(\boldsymbol{\theta}^i)$ or $\mathcal{L}_{\mathcal{D}^l}(\boldsymbol{\theta}^i) < \mathcal{L}_{\mathcal{D}^l}(\boldsymbol{\theta}) < \mathcal{L}_{\mathcal{D}^l}(\boldsymbol{\theta}^d)$.*

**Proof 1** *To simplify the problem, from a stochastic perspective with one instance $(u, i, y_{ui}^*)$. Our target is to satisfy:*

$$\mathcal{L}(\boldsymbol{\theta}^d) > \mathcal{L}(\boldsymbol{\theta}) > \mathcal{L}(\boldsymbol{\theta}^i), \tag{5}$$

*or*

$$\mathcal{L}(\boldsymbol{\theta}^d) < \mathcal{L}(\boldsymbol{\theta}) < \mathcal{L}(\boldsymbol{\theta}^i), \tag{6}$$

*where $\mathcal{L}(\boldsymbol{\theta}^d), \mathcal{L}(\boldsymbol{\theta}), \mathcal{L}(\boldsymbol{\theta}^i)$ are the loss functions for the models with parameters $\boldsymbol{\theta}^d, \boldsymbol{\theta}, \boldsymbol{\theta}^i$. To simplify the proof procedure, we define the prediction function as the hypothesis function of the well known logistic regression (Hosmer Jr et al., 2013)[1]:*

$$\hat{y}_{ui}^{\boldsymbol{\theta}} = f(u, i|\boldsymbol{\theta}) = f(\boldsymbol{x}_{u,i}|\boldsymbol{\theta}) = \frac{1}{1 + e^{-\boldsymbol{\theta} \boldsymbol{x}_{u,i}}}, \tag{7}$$

*where $\boldsymbol{x}_{u,i}$ is the input feature under the interaction between user $u$ and item $i$. Given learning rate $\alpha \in \mathbb{R}, \alpha \neq 0$ with $\boldsymbol{\theta}^d = \boldsymbol{\theta} - \alpha \circ \nabla \mathcal{L}_{\mathcal{D}^u}^{dual}(\boldsymbol{\theta})$ and $\boldsymbol{\theta}^i = \boldsymbol{\theta} + \alpha \circ \nabla \mathcal{L}_{\mathcal{D}^u}^{dual}(\boldsymbol{\theta})$ as the temporal model parameters updated by the direct gradient and inverse gradient, respectively, we can have:*

$$\boldsymbol{\theta}^d \boldsymbol{x}_{u,i} = \boldsymbol{\theta} \boldsymbol{x}_{u,i} - \alpha \circ \nabla \mathcal{L}_{\mathcal{D}^u}^{dual}(\boldsymbol{\theta}) \boldsymbol{x}_{u,i} \tag{8}$$

$$\boldsymbol{\theta}^i \boldsymbol{x}_{u,i} = \boldsymbol{\theta} \boldsymbol{x}_{u,i} + \alpha \circ \nabla \mathcal{L}_{\mathcal{D}^u}^{dual}(\boldsymbol{\theta}) \boldsymbol{x}_{u,i} \tag{9}$$

*Assume $\alpha \circ \nabla \mathcal{L}_{\mathcal{D}^u}^{dual}(\boldsymbol{\theta}) \boldsymbol{x}_{u,i} < 0$ then we have:*

$$\boldsymbol{\theta}^d \boldsymbol{x}_{u,i} > \boldsymbol{\theta} \boldsymbol{x}_{u,i} > \boldsymbol{\theta}^i \boldsymbol{x}_{u,i} \tag{10}$$

*Review the loss function from a stochastic perspective with one instance $(u, i, y_{ui}^*)$:*

$$\mathcal{L}(\boldsymbol{\theta}) = -y_{ui}^* \log \left( \hat{y}_{ui}^{\boldsymbol{\theta}} \right) - (1 - y_{ui}^*) \log \left( 1 - \hat{y}_{ui}^{\boldsymbol{\theta}} \right). \tag{11}$$

*Suppose $(u, i, y_{ui}^*)$ is positive instance with $y_{ui}^* = 1$, then we can have:*

$$\mathcal{L}(\boldsymbol{\theta}) = -\log \left( \hat{y}_{ui}^{\boldsymbol{\theta}} \right), \tag{12}$$

*which is a decreasing function towards predicted probability $\hat{y}_{ui}^{\boldsymbol{\theta}}$.*

*$\mathcal{L}(\boldsymbol{\theta})$ will decrease towards $\hat{y}_{ui}^{\boldsymbol{\theta}}$ and further decrease towards $\boldsymbol{\theta} \boldsymbol{x}_{u,i}$. Base on the assumption of Eqn.equation 10, we can have:*

$$\mathcal{L}(\boldsymbol{\theta}^d) < \mathcal{L}(\boldsymbol{\theta}) < \mathcal{L}(\boldsymbol{\theta}^i), \tag{13}$$

*which satisfies the target of Eqn.equation 6. Besides, similarly, if $y_{ui}^* = 0$, we can have: $\mathcal{L}(\boldsymbol{\theta}^d) > \mathcal{L}(\boldsymbol{\theta}) > \mathcal{L}(\boldsymbol{\theta}^i)$. If $\alpha \circ \nabla \mathcal{L}_{\mathcal{D}^u}^{dual}(\boldsymbol{\theta}) \boldsymbol{x}_{u,i} > 0$, we can have similar conclusion. That is to say, whatever cases, targets of Eqn.equation 5 and Eqn.equation 6 will be satisfied.*

*The case $\alpha \circ \nabla \mathcal{L}_{\mathcal{D}^u}^{dual}(\boldsymbol{\theta}) \boldsymbol{x}_{u,i} = 0$ will result in gradient vanishing and modern machine learning approach often randomly initialize the features to avoid such case.*

## A.2 ALGORITHM

The algorithm of our proposed inverse gradient is as Algorithm 1, where labeled data $\mathcal{D}^l$, unlabeled data $\mathcal{D}^u$ and learning rate $\gamma, \alpha$ are required as input. The first iteration aims to pre-train the model using the split training-train data. The second iteration aims to explore the direct gradient and inverse gradient on the loss for sampled unlabeled data, where three strategies are explored here as line 13-15 with $\boldsymbol{\theta}^d, \boldsymbol{\theta}^i$ and $\boldsymbol{\theta}$ as the model parameters updated by the direct gradient, inverse gradient and without being updated by the gradient on the loss for sampled unlabeled data, respectively. Finally, the explored updated direction with minimal test loss on the training-test data will be selected to update the model for this iteration.

---

[1] https://www.coursera.org/learn/machine-learning

---

**Algorithm 1:** Inverse Gradient Adaptation

---

**input** : Labeled data $\mathcal{D}^l$, unlabeled data $\mathcal{D}^u$, learning rate $\gamma$, $\alpha$
**output:** $\theta$

1   *Initialize $\theta$, split $\mathcal{D}^l$ into training-train data* $\text{train}\left(\mathcal{D}^l\right)$ *and training-test data* $\text{test}\left(\mathcal{D}^l\right)$;
2   **while** *not done* **do**
3     **for** $t = 1$ **to** $T$ **do**
4       *Sample batch of training labeled data* $\text{train}(\mathcal{D}_t^l) \sim \mathcal{D}^l$;
5       $\mathcal{L}_{\text{train}\left(\mathcal{D}_t^l\right)}(\theta) = \frac{1}{\left|\text{train}\left(\mathcal{D}_t^l\right)\right|} \sum_{(u,i,y_{ui}^*)\in\text{train}\left(\mathcal{D}_t^l\right)} \ell\left(\hat{y}_{ui}^{\theta}, \hat{y}_{ui}^*\right)$;
6       $\theta = \theta - \gamma \circ \nabla \mathcal{L}_{\text{train}(\mathcal{D}_t)}(\theta)$;
7     **for** $t = 1$ **to** $T$ **do**
8       *Sample batch of training unlabeled data* $\text{train}(\mathcal{D}_t^u) \sim \mathcal{D}^u$;
9       $\mathcal{L}_{\text{train}\left(\mathcal{D}_t^u\right)}^{dual}(\theta) = \frac{1}{\left|\text{train}\left(\mathcal{D}_t^u\right)\right|} \sum_{(u,i,\bar{y}_{ui})\in\text{train}\left(\mathcal{D}_t^u\right)} w^1 \ell\left(\hat{y}_{ui}^{\theta}, 1\right) + w^0 \ell\left(\hat{y}_{ui}^{\theta}, 0\right)$;
10      $\theta^d = \theta - \alpha \circ \nabla \mathcal{L}_{\text{train}\left(\mathcal{D}_t^u\right)}^{dual}(\theta)$;
11      $\theta^i = \theta + \alpha \circ \nabla \mathcal{L}_{\text{train}\left(\mathcal{D}_t^u\right)}^{dual}(\theta)$;
12      *Sample batch of test labeled data* $\text{test}(\mathcal{D}_t^l) \sim \mathcal{D}^l$;
13      $\mathcal{L}_{\text{test}\left(\mathcal{D}_t^l\right)}(\theta^d) = \frac{1}{\left|\text{test}\left(\mathcal{D}_t^l\right)\right|} \sum_{(u,i,y_{ui}^*)\in\text{test}\left(\mathcal{D}_t^l\right)} \ell\left(\hat{y}_{ui}^{\theta^d}, y_{ui}^*\right)$;
14      $\mathcal{L}_{\text{test}\left(\mathcal{D}_t^l\right)}(\theta^i) = \frac{1}{\left|\text{test}\left(\mathcal{D}_t^l\right)\right|} \sum_{(u,i,y_{ui}^*)\in\text{test}\left(\mathcal{D}_t^l\right)} \ell\left(\hat{y}_{ui}^{\theta^i}, y_{ui}^*\right)$;
15      $\mathcal{L}_{\text{test}\left(\mathcal{D}_t^l\right)}(\theta) = \frac{1}{\left|\text{test}\left(\mathcal{D}_t^l\right)\right|} \sum_{(u,i,y_{ui}^*)\in\text{test}\left(\mathcal{D}_t^l\right)} \ell\left(\hat{y}_{ui}^{\theta}, y_{ui}^*\right)$;
16      $\theta = \arg\min_{\{\theta^d, \theta, \theta^i\}} \{\mathcal{L}_{\text{test}\left(\mathcal{D}_t^l\right)}(\theta^d), \mathcal{L}_{\text{test}\left(\mathcal{D}_t^l\right)}(\theta), \mathcal{L}_{\text{test}\left(\mathcal{D}_t^l\right)}(\theta^i)\}$;
17      $\mathcal{L}_{\text{test}\left(\mathcal{D}_t^l\right)}(\theta) = \min\{\mathcal{L}_{\text{test}\left(\mathcal{D}_t^l\right)}(\theta^d), \mathcal{L}_{\text{test}\left(\mathcal{D}_t^l\right)}(\theta), \mathcal{L}_{\text{test}\left(\mathcal{D}_t^l\right)}(\theta^i)\}$;
18      $\theta = \theta - \gamma \circ \nabla \mathcal{L}_{\text{test}\left(\mathcal{D}_t^l\right)}(\theta)$;

---

### A.3   DATASETS AND PRE-PROCESSING

Table 2: Data statistics for processed Micro-Video dataset and ML1M dataset.

| Dataset | Users | Items | Feedback | | | Density |
| --- | --- | --- | --- | --- | --- | --- |
| | | | Positive | Negative | Total | |
| **Micro-Video** | 37,692 | 131,690 | 4,915,745 | 4,546,747 | 9,462,492 | 0.19% |
| **ML1M** | 6,041 | 3,953 | 836,478 | 163,731 | 1,000,209 | 4.19% |

The public ML1M dataset is published at[2] and we also have uploaded the processed dataset on the Github of the code and the supplementary material. The statistics of our adopted Micro-Video dataset and ML1M dataset are as Table 2. We will put the Micro-Video dataset public to benefit the community.

Table 3: Interaction statistics for Micro-Video dataset.

| Dataset | Like | Hate | Total | Like/Total | Hate/Total |
| --- | --- | --- | --- | --- | --- |
| **Micro-Video** | 382,570 | 6,157 | 8,008,965 | 4.78% | 0.08% |

**Micro-Video** This dataset is collected from one of the largest Micro-Video platforms in China, where user behaviors such as playing time, like and hate are recorded. The data is downsampled from September 11 to September 22, 2021. Users are passive to receive the recommended videos here, and there are extremely limited active feedback such as like and hate as shown in Table 3. That is to say, we have extremely limited reliable feedback in this data which is very challenging in modern industries. We take the active feedback of like and hate to analyse the regular pattern of playing time and duration of each video which are included in each interaction. From Figure 6, we can discover that the users' like and hate behaviors are related to the finish rate of playing time (user's

---

[2]https://grouplens.org/datasets/movielens/1m/

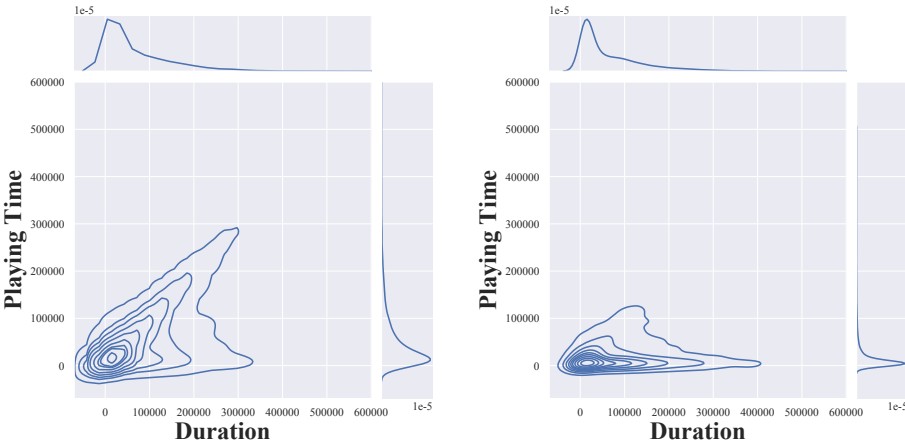

Figure 6: Joint distribution of playing time and duration for like (left) and hate (right).

Table 4: Performance comparison of GMF on Micro-Video dataset based on feedback classification by finish rate and playing time.

| metric | AUC | GAUC |
|---|---|---|
| Finish Rate | 0.8024 | 0.7558 |
| Playing Time | 0.751 | 0.528 |

playing time towards certain item) to be divided by duration (item's total time): when the users like the video, they are more likely to finish watching it, and vice versa. Thus we treat the finish rate greater than 80% and less than 20% as positive and negative feedback, respectively. Another way to classify the positive and negative feedback is according to the playing time, and we also report the results by treating the playing time over upper quartile and under lower quartile as positive and negative feedback, respectively. From Table 4, we can observe that finish rate is more suitable for pattern capturing and we adopt such way as feedback processing in experimental evaluation.

**ML1M**[3] This is a widely used public movie dataset in recommendation. The rating score in ML1M ranging from 1 to 5 and we treat the rating score over 3 and under 2 as positive and negative feedback, respectively, following DenoisingRec (Wang et al., 2021).

Besides, we splite 60%, 20% and 20% of the data as training, validation and test data, for these two datasets.

### A.4    IMPLEMENTATION DETAILS

All the models are implemented based on Python with a Pytorch[4] framework based on the repository DenoisingRec[5]. The environment is as below.

- Anaconda 3

- python 3.7.3

- pytorch 1.4.0

- numpy 1.16.4

---

[3]https://grouplens.org/datasets/movielens/1m/
[4]https://pytorch.org/
[5]https://github.com/WenjieWWJ/DenoisingRec

### A.5 HYPER-PARAMETER STUDY

**Impact of learning rate for convergence** We vary the learning rate for inverse dual loss with 1, 10, 50 and 100 times as much as the learning rate for test loss, to investigate the convergence analysis at Section 2.3.2. From Figure 7, we can discover that the greater of $\alpha$ will truly result in gradient ascent, as there appears pass gradient when $\alpha = 100\gamma$. Besides, the greater of $\alpha$, the faster of convergence on training-test data but also overfitting.

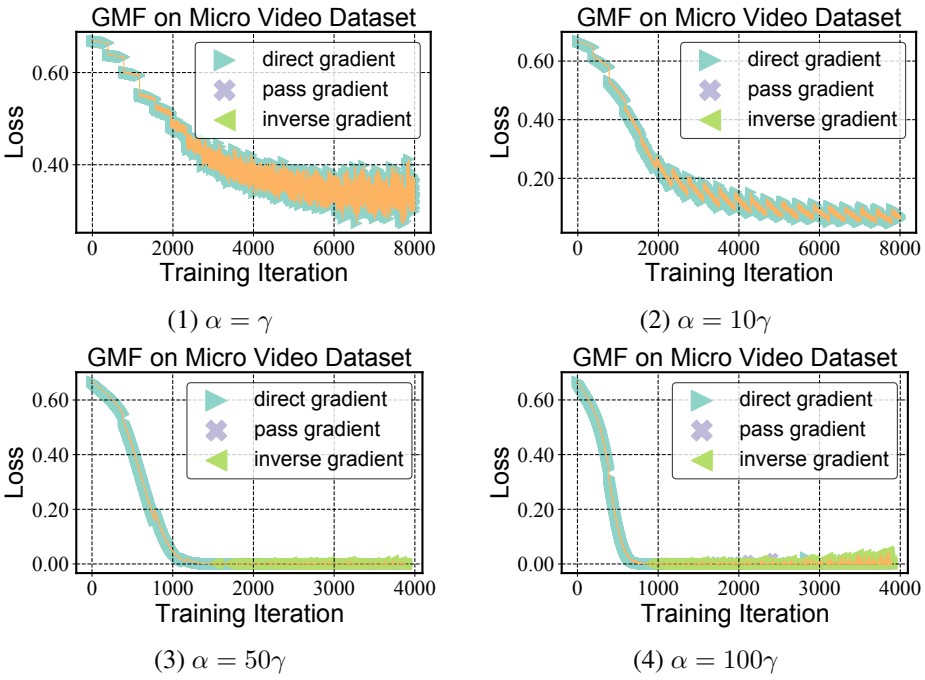

Figure 7: The loss on training-test data with different learning rate for adapted gradient.

### A.6 COMPARISON WITH HARD NEGATIVE SAMPLING

Table 5: Performance comparisons towards hard negative sampling with GMF and NeuMF backbones on two datasets. Bold and underline refer to the best and second best result, respectively

| Model | Method | ML1M | | | Micro Video | | |
|---|---|---|---|---|---|---|---|
| | | NDCG | AUC | GAUC | NDCG | AUC | GAUC |
| GMF | None | 0.9285 | 0.7671 | 0.6919 | 0.7365 | 0.8024 | 0.7558 |
| | NS | 0.9400 | 0.7639 | 0.7200 | 0.7049 | 0.7802 | 0.7247 |
| | DNS | 0.9178 | 0.6290 | 0.6290 | 0.6878 | 0.6310 | 0.6356 |
| | SRNS | 0.9176 | 0.6253 | 0.6293 | 0.6799 | 0.6293 | 0.6302 |
| | IDL | 0.8996 | 0.7304 | 0.6400 | 0.7272 | 0.7858 | 0.7335 |
| | IG | **0.9521** | **0.8318** | **0.7642** | **0.7773** | **0.8033** | **0.7593** |
| NeuMF | None | 0.9214 | 0.7524 | 0.6856 | 0.6649 | 0.7504 | 0.6916 |
| | NS | 0.9298 | 0.7495 | 0.7088 | 0.6350 | 0.7191 | 0.6689 |
| | DNS | 0.9168 | 0.6277 | 0.6283 | 0.6203 | 0.6274 | 0.6261 |
| | SRNS | 0.9172 | 0.6282 | 0.6284 | 0.6212 | 0.6296 | 0.6295 |
| | IDL | 0.9212 | 0.7962 | 0.6984 | 0.6466 | 0.7174 | 0.6647 |
| | IG | **0.9449** | **0.8253** | **0.7569** | **0.7809** | **0.8198** | **0.7689** |

The results of performance comparison with hard negative sampling are as Table 5, where we can discover that the hard negative sampling methods truly perform worse in our setting where both

ground-truth positive and negative samples are tested. The typical and state-of-the-art baselines of hard negative sampling are illustrated as below.

- DNS (Zhang et al., 2013): Dynamic Negative Sampling (DNS) samples unlabeled instances as negatives and picks the hard instances with high predicted score.
- SRNS (Ding et al., 2020): SRNS further selects true negatives with high-variance to improve DNS.

