# OpenReview forum: "Inverse Learning with Extremely Sparse Feedback for Recommendation"
_ICLR.cc/2023/Conference — Submitted to ICLR 2023_

### Official Review · Reviewer_a7qD · 2022-10-24

**Confidence:** 4
**Correctness:** 3
**Technical Novelty And Significance:** 2
**Empirical Novelty And Significance:** 2
**Recommendation:** 5

**Clarity, Quality, Novelty And Reproducibility:**

Clarity is good and easy to follow.

Quality is fair.

Novelty is fair that the idea is not surprising.

Reproducibility is good since code is/will be public according to the main text.

**Strength And Weaknesses:**

Strength
The paper is clear to follow. The motivation and intuition is clear and valid.

Weakness
There is no rigorous theoretical guarantee. There is no data point in a complicated experimental set up such as industrial level production A/B testing results.


**Summary Of The Paper:**

The paper introduces a heuristic method to better exploit unlabeled data in recommender system in order to boost performance and also avoid too strong introduced noise. The method has components such as a re-weighted loss based on current model prediction and an explorative gradient selection logic. The performance gain demonstrates the necessarily of the gradient selection logic.

**Summary Of The Review:**

The author should also benchmark with semi-supervised learning in recommender system since both trying to exploit (assign a label) to large volume unlabeled data.

When model is not well trained (such as in early stage training), there will be a negative feedback loop to use such inverse loss and inverse gradient. The method is too heuristic to cope with such edge cases.

Flaws:
Usually the predicted label is between 0 and 1 and the loss in Eq one is always positive. In Eq 4, the absolute symbol doesn't make sense.

The inverse gradient is too ad hoc. For example, in the proof, we see a very much simplified case, this is not sufficient proof for a theorem. It's very hard to define easy example, hard example and guarantee the gradient triage to work.

Typo:
in abstract line 3, it should be "often introduces false-negative noises", instead of "often introduces false-positive noises"?

---

> ### Author Response · Authors · 2022-11-16
> **Response to Reviewer a7qD**
>
> Thanks for your valuable comments, and we hope the following responses can address your concerns.
>
> **Q1. The author should also benchmark with semi-supervised learning in recommender system since both try to exploit (assign a label) a large volume of unlabeled data.**
>
> **Response:** Thanks for your valuable suggestion. Indeed our work follows denoising recommender systems which achieve denoising on the noisy label by weighting or truncating the loss [Wenjie Wang et al. 2021]. We do not explicitly assign a label to each instance like semi-supervised learning. Instead, we weigh the loss of each instance from both positive and negative labels.
>
>
> [Wenjie Wang et al. 2021] Denoising implicit feedback for recommendation, in WSDM 2021.
>
>
> **Q2. When the model is not well trained (such as in early stage training), there will be a negative feedback loop to use such inverse loss and inverse gradient. The method is too heuristic to cope with such edge cases.**
>
> **Response:** Thanks for your valuable suggestion. In early stage training, the model will be well pre-trained by the training-train data in our Algorithm 1 of Appendix A.2. Besides, the loss descents of Figure 5 and Figure 7 have illustrated that our method indeed can cope with such edge cases.
>
>
>
> **Q3. Usually the predicted label is between 0 and 1 and the loss in Eq one is always positive. In Eq 4, the absolute symbol doesn't make sense.**
>
> **Response:** The predicted label is truly between 0 and 1, and the loss in Eq one is always positive. However, in Eq 4, the absolute symbol indeed makes sense. Though the weights in Eq 4 are between 0 and 1 and are always positive, we have both positive weight and negative weight for the positive label and negative label, respectively. When the positive weight is greater than the negative weight, the instance will be classified as positive and vice versa. Besides, the results of Figure 3 and Figure 4 can well show the effectiveness of our proposed method to annotate the positive and negative labels.
>
>
> **Q4. The inverse gradient is too ad hoc. For example, in the proof, we see a very simplified case. This is not sufficient proof for a theorem. It's very hard to define easy example, hard example and guarantee the gradient triage to work.**
>
> **Response:** Thanks for your valuable suggestion. To illustrate the proof procedure and roundly support the theorem, we exploit both the simplified case and experimental results.
>
> Though it's very hard to guarantee the gradient triage to work in a purely theoretical way, we can still prove it by the results in Figure 5, where the loss is always updated by either direct or inverse gradients.
>
> **Q5. In abstract line 3, it should be "often introduces false-negative noises", instead of "often introduces false-positive noises"?**
>
> **Response:** Thanks for your valuable suggestion. We are sorry for the typo, and it should be "false-negative noises".

---

### Official Review · Reviewer_6jCU · 2022-10-26

**Confidence:** 4
**Clarity, Quality, Novelty And Reproducibility:** 1.	The motivation to develop the inve…
**Correctness:** 3
**Technical Novelty And Significance:** 3
**Empirical Novelty And Significance:** 3
**Recommendation:** 3

**Strength And Weaknesses:**

Strengths:
1.	The authors propose to perform denoising from both positive and negative perspectives in recommendation.
2.	The authors propose two new methods, i.e., inverse dual loss and inverse gradient, to improve the model performance.
3.	The authors perform extensive experiments on two real datasets to demonstrate the effectiveness of the proposed method.

Weaknesses:
1.	The proposed method is only studied with the logloss. It is not clear whether the proposed method can help other ranking-based recommendation loss, e.g., Bayesian personalized ranking loss.
2.	The backbone models, i.e., GMF and NeuMF, are a little weak. The authors are recommended to consider other advanced recommendation models as backbone models, e.g., LightGCN and other sequential recommendation models.
3.	Some claims in this paper are not supported by experimental analysis.


**Summary Of The Paper:**

This paper studies the de-noising recommendation problem. The authors propose a meta learning method to annotate the unlabelled data from both the loss and gradient perspective. They propose the inverse dual loss to boost the true label learning and prevent false label learning. Moreover, they also propose the inverse gradient to explore the correct updating gradient. To demonstrate the effectiveness of the proposed method, the authors perform extensive experiments on two real datasets.

**Summary Of The Review:**

Overall, this paper provides some novel ideas to solve the drawbacks of existing negative sampling strategies widely used in existing recommendation research. Some claims in this paper are not supported by experiments. Moreover, the experimental analysis is not sufficient. The authors only study the effectiveness of the proposed method on the Logloss. The quality of this work can be improved by studying the performance of the proposed method with other recommendation functions, e.g., BPR loss. In addition, the backbone models are a little weak. The authors need to consider other advanced recommendation models, e.g., LightGCN or other sequential recommendation models, to demonstrate the effectiveness of the proposed method.

---

> ### Author Response · Authors · 2022-11-16
> **Response to Reviewer 6jCU (part 2)**
>
>
>
> **Q4. In Section 2.2.1, the authors claim that the objective of this work is to propose a noiseless solution for unlabeled data. However, the definition of “noiseless” is not clear. Moreover, there is no experimental analysis studying the “noiseless” property of the proposed method.**
>
> **Response:** Thanks for your valuable question. We are sorry for our improper usage of word here, and the *noiseless* here should be *denoising*.
>
>
>
> **Q5. Why choose the Micro-video dataset for experiments instead of other publicly available datasets, e.g., the Amazon Review dataset?**
>
> **Response:** Thanks for your valuable question. This is because, in the Micro-video dataset, users are passive in receiving the recommended items without any active clicking or rating action. That is to say, we have a large number of unlabeled feedback with extremely sparse labeled data, which is a key challenge in this work. We have described this challenge of the video feed recommendation problem in the first paragraph of the introduction.
>
>
> **Q6. In the experiments, the definition of positive and negative feedback is based on some pre-defined rules, e.g., a finishing rate greater than 80% as positive feedback and less than 20% as negative feedback. Are these kinds of definitions reasonable.**
>
> **Response:** We have visualized the distribution of playing time and duration for like (positive feedback) and hate (negative feedback) in Figure 6 of Appendix A.3, where the results show that the finishing rate is more suitable to classify them. Pre-defined rules of greater than 80% as positive feedback and less than 20% as negative feedback are not the focuses of our work, but it is necessary for us to keep the settings for training and testing consistent.
>
>
>
>
> **Q7. In the experiments, the authors claimed that “we split 60%, 20%, and 20% of the data as training, validation, and test data, for these two datasets”. However, it is not clear whether this data splitting is based on each individual user’s behavior or all users’ behavior data.**
>
> **Response:** Thanks for your valuable question. We are sorry for our ambiguous description of data splitting. We indeed split the data based on all users’ behavior data.

---

> ### Author Response · Authors · 2022-11-16
> **Response to Reviewer 6jCU (part 1)**
>
> Thanks for your valuable comments, and we hope the following responses can address your concerns.
>
> **Q1. The proposed method is only studied with the logloss. It is not clear whether the proposed method can help other ranking-based recommendation loss, e.g., Bayesian personalized ranking loss.**
>
> **Response:** Our proposed method does not depend on specific loss, so we leverage the widely used logloss here[Guorui Zhou et al. 2019; Guorui Zhou et al. 2018].
> Bayesian personalized ranking loss can also replace the logloss here because we can sample positive and negative instances to construct the negative loss and positive loss, respectively, for our inverse dual loss.
>
>
> [Guorui Zhou et al. 2019] Deep interest evolution network for click-through rate prediction, in AAAI 2019.
>
> [Guorui Zhou et al. 2018] Deep interest network for click-through rate prediction, in KDD 2018.
>
>
> **Q2. The backbone models, i.e., GMF and NeuMF, are a little weak. The authors are recommended to consider other advanced recommendation models as backbone models, e.g., LightGCN and other sequential recommendation models.**
>
> **Response:**  Here, we focus on the denoising method and compare it with baselines based on backbones from existing work [Wenjie Wang et al. 2021]. Indeed our task is different from the sequential recommendation, which models the dynamic interest of users and requires timestamp information for each interaction [Hui Fang et al. 2021].
>
> [Wenjie Wang et al. 2021] Denoising implicit feedback for recommendation, in WSDM 2021.
> [Hui Fang et al. 2021] Deep Learning for Sequential Recommendation: Algorithms, Influential Factors, and Evaluations, in ACM TOIS 2021.
>
> **Q3. The motivation for developing the inverse gradient method is to achieve more robust sampling on hard instances. However, there is no experimental analysis studying the “robustness” of sampling on hard instances.**
>
> **Response:** Thanks for your insightful question. We would like to verify the robustness of sampling by comparing the inverse gradient (IG) with the inverse dual loss (IDL, w/o inverse gradient) on two different sampled data, as shown in the table below. Here diff% means difference. From the table below, we can observe that after removing the inverse gradient, the inverse dual loss is more sensitive to the sampled data (i.e. up to 0.8% fluctuation with the different sampled data). That is to say, equipping inverse gradient will be more robust.
>
> |    **Component**   |   **IDL (w/o IG)**| **IDL (w/o IG)** | **IDL (w/o IG)**  |  **IG**  |  **IG** | **IG**   |
> |:------------------:|:--------:|:-------:|:--------:|:--------:|:-------:|:--------:|
> |     **Metric**     | **NDCG** | **AUC** | **GAUC** | **NDCG** | **AUC** | **GAUC** |
> | **Sampled data A** |  0.8996  | 0.7304  |  0.6400  |  0.9521  | 0.8318  |  0.7642  |
> | **Sampled data B** |  0.9043  | 0.7364  |  0.6451  |  0.9522  | 0.8318  |  0.7645  |
> |      **diff%**     |   0.51%  |  0.82%  |   0.80%  |  -0.01%  |  0.00%  |  -0.04%  |

---

### Official Review · Reviewer_16D4 · 2022-10-31

**Confidence:** 4
**Correctness:** 2
**Technical Novelty And Significance:** 2
**Empirical Novelty And Significance:** 2
**Recommendation:** 5

**Clarity, Quality, Novelty And Reproducibility:**

- Clarity is lacking in certain parts of the paper (see concerns).
- Quality of the work is not up to the standard of acceptance as several parts cause serious confusion and the experiments are not comprehensive (see concerns).
- The method proposed in this work is considered quite novel in my opinion, especially the inverse dual loss and the application of the meta-learning framework.
- The work is reproducible as both the source code and the datasets used are released.


**Strength And Weaknesses:**

Strengths
- The paper focuses on solving an important and challenging issue of noisy negative sampling in recommender systems.
- The idea of employing the inverse dual loss to annotate the unlabeled data and applying a meta-learning framework to determine the update gradient for more robust learning is novel.
- A processed Micro-Video dataset is released to the public, where labeling has been done based on the playing time of videos. This I believe will become a good contribution to the community.
- The experiments to analyze the effectiveness of IDL and IG are helpful.

Weaknesses (see the concerns below)
- Some claims in the paper are not well-supported.
- Several parts of the paper lack clarity, which affects the understanding of some important ideas.
- Experiments can be further improved to include the implicit feedback setting.

Major Concerns
1. In section 2.3.1, the authors claim that "when the sampled data is hard, exploiting the inverse gradient to update the model will gain a smaller test loss on the training-test data, and thus we exploit inverse gradient here". However, I find that the link between the hard instances and the inverse gradient is not clear, and hence the claim that the inverse gradient helps improve learning with the hard instances is not supported.\
Based on the descriptions in the previous sections (and illustrations in Fig. 1), hard instances are points that are close to the decision boundary (and easy instances are those that are far from the decision boundary). This means that for these hard unlabeled instances, the negative loss and positive loss values are close to each other (as the classifier has a hard time determining labels for them), which results in similar weights for the negative and positive loss in the inverse dual loss in eq. 4. Hence, for hard instances, the resulting effect on learning is just that the model will hardly change as both the positive and negative loss are optimized equally. I don't see why in this case, employing an inverse gradient of the inverse dual loss will help achieve better learning and result in "a smaller test loss on the training-test data" (similarly for the case of easy instances, why employing direct gradient for them is better). The authors may want to provide more explanations on this to support the claim.
2.  The experiments in this paper analyze two scenarios: passive feed recommendations (Micro-Video) and explicit user rating (ML1M). For both cases, the authors pre-processed the datasets to have binary labeling where the labeled user-item pairs can be either positive or negative. However, for the implicit feedback setting (i.e., users provide interactions like clicks and purchases as feedback), all the labeled/interacted user-item pairs are considered positive. For this case, the test loss in the meta-learning procedure will only have positive data points. How will this affect the annotation quality of the inverse dual loss for the unlabeled/uninteracted data? Since the implicit setting is quite a prevalent scenario in the industry, I suggest the authors incorporate experiments on the implicit setting (e.g., using Yelp, Amazon datasets) to make the evaluation of the proposed method more comprehensive.

Minor Concerns
1. In Definition 1, it is not clear why the normalization parameter z_w is formulated in this way. Can the authors provide some intuitions on this?
2. In the summary of the main contributions, the authors claim that they are "the first to perform denoising from both positive and negative perspectives in recommendation". According to my understanding, the paper only proposes a method to denoise the negative sampling (avoid false-negative) from the unlabeled data. Can the authors explain why they claim that their method also serves to perform denoising from a positive perspective?
3. There is this recent work [1] that aims to tackle noises in both positive and negative implicit feedback. The authors may consider including a comparison with this work (adding it as a baseline in experiments or at least discussing it in the related work).

Parts that need further clarification/adjustment
1. Based on the descriptions in "Baselines and Backbones" and section 3.1, it is not clear whether IG is adding on top of IDL or the two methods are applied separately.
2. In the second paragraph of section 3.1, what does it mean by "IDL only outperforms the NeuMF" and "it is even outperformed by GMF", as both NeuMF and GMF are just the backbones but not the methods to compare? The authors may want to rephrase this part.
3. In section 3.3, the authors state something about the effects of pre-training on the training-train data. However, experiments in this section all involve pre-training, and a set of experiments without pre-training is required to make comparisons and support such claims.
4. In Fig. 5, for the two plots on NeuMF, the legend boxes are too large and block out some important parts of the learning curves.

[1] Denoising Neural Network for News Recommendation with Positive and Negative Implicit Feedback, NAACL 2022


**Summary Of The Paper:**

To tackle the noisy negative sampling from the unlabeled data, the paper proposes an inverse dual loss (IDL) to assign larger weights to the more confident loss (either negative loss or positive loss) on the unlabeled data to achieve automatic labeling. To further improve learning robustness, it proposes a meta-learning procedure to determine whether a direct gradient, an inverse gradient (IG), or a pass gradient of the inverse dual loss will be applied based on the meta-test performance. Experiments on two datasets demonstrate the method's superiority over naive negative sampling and a denoising method that only focuses on denoising the labeled data. The effectiveness of IDL to classify the unlabeled data and the effectiveness of IG to assist convergence are also analyzed in the experiments.

**Summary Of The Review:**

Since some important claims made in the paper are ambiguous/not well-supported and the experiments lack evaluation in an important aspect (i.e., implicit setting), I feel that the paper is marginally below the acceptance threshold.

---

> ### Author Response · Authors · 2022-11-16
> **Response to Reviewer 16D4 (part 2)**
>
>
> **Q5. There is this recent work [1] that aims to tackle noises in both positive and negative implicit feedback. The authors may consider including a comparison with this work.**
>
> **Response:** Thanks for your valuable suggestion. The work you mentioned [1] is about news recommendation where additional text information is exploited to achieve denoising. Similar to us, they also consider both positive and negative perspectives. However, it highly relies on additional news text information, and thus they cannot be considered as baselines.
>
> **Q6. Based on the descriptions in "Baselines and Backbones" and section 3.1, it is not clear whether IG is adding on top of IDL or the two methods are applied separately.**
>
> **Response:** Thanks for your valuable question. IG is actually adding on the top of IDL which has been illustrated in Algorithm 1 of appendix A.2.
>
>
> **Q7. In the second paragraph of section 3.1, what does it mean by "IDL only outperforms the NeuMF" and "it is even outperformed by GMF", as both NeuMF and GMF are just the backbones but not the methods to compare?**
>
> **Response:** Thanks for your valuable question. We are sorry for this ambiguous description. Here NeuMF and GMF refer to the backbone without any samlping or denoising methods, such as the None method in Table 1 (w.r.t. None in Table 1).
>
>
> **Q8. In section 3.3, the authors state something about the effects of pre-training on the training-train data. However, experiments in this section all involve pre-training, and a set of experiments without pre-training is required to make comparisons and support such claims.**
>
> **Response:** Thanks for your valuable question. We have experimented on the setting without pre-training and the results are much more poor than the pre-training setting. We put the results as below, from which we can observe that the method with pre-training even outperform the method without pre-training by around 5.94% on Micro Video dataset under AUC.
>
> | Model |      Method      |   ML1M  |     ML1M    |  ML1M       | Micro Video |    Micro Video     |   Micro Video      |
> |:-----:|:----------------:|:-------:|:-------:|:-------:|:-----------:|:-------:|:-------:|
> |       |                  |   NDCG  |   AUC   |   GAUC  |     NDCG    |   AUC   |   GAUC  |
> |  GMF  | w/o pre-training | 0.9397  | 0.7559  | 0.7267  |   0.7151    | 0.7582  | 0.7151  |
> |    GMF   |  w pre-training  | 0.9521  | 0.8318  | 0.7642  |   0.7773    | 0.8033  | 0.7593  |
> | NeuMF | w/o pre-training | 0.9411  | 0.8040  | 0.7264  |   0.7398    | 0.7882  | 0.7543  |
> |    NeuMF   |  w pre-training  | 0.9449  | 0.8253  | 0.7569  |   0.7809    | 0.8198  | 0.7689  |
>
>
> **Q9. In Fig. 5, for the two plots on NeuMF, the legend boxes are too large and block out some important parts of the learning curves.**
>
> **Response:** Thanks for your valuable suggestion. We will tune the legend boxes to clarify the important parts in the revised version.

---

> ### Author Response · Authors · 2022-11-16
> **Response to Reviewer 16D4 (part 1)**
>
> Thanks for your valuable comments, and we hope the following responses can address your concerns.
>
> **Q1. For hard instances, the resulting effect on learning is just that the model will hardly change as both the positive and negative loss are optimized equally. In this case, why employing an inverse gradient of the inverse dual loss will help achieve better learning?**
>
> **Response:**  Thanks for your valuable question. For hard instances around the classification boundary, the model actually will hardly change as both the positive and negative loss are optimized equally. We are sorry for the ambiguous description of this case. The hard instances here indeed refers to the hard instances that are misclassified by the model as Figure 1 (j) and (l), instead of the instances that are around the decision boundary.
>
>
> **Q2. For the implicit feedback setting (i.e., users provide interactions like clicks and purchases as feedback), all the labeled/interacted user-item pairs are considered positive. For this case, the test loss in the meta-learning procedure will only have positive data points. How will this affect the annotation quality of the inverse dual loss for the unlabeled/uninteracted data?**
>
>
> **Response:** Here we tend to tackle the recommendation problem where both positive feedback and negative feedback are provided such as video feed recommendation. In the traidional problem of learning from implicit feedback, the negative data points should be sampled (unconvinced test loss) and thus the problem may not be suitable to validate the effectiveness of our proposed model.
>
>
> **Q3. In Definition 1, it is not clear why the normalization parameter $z_w$ is formulated in this way. Can the authors provide some intuitions on this?**
>
> **Response:** It is a normalization term.
> $\\frac{\|\\ell\\left(\\hat{y}^{\\boldsymbol{\\theta}}\_{u i}, 0\\right)\|}{\|\\ell\\left(\\hat{y}^{\\boldsymbol{\\theta}}\_{u i}, 1\\right)\|}$
> and
> $\\frac{\|\\ell\\left(\\hat{y}^{\\boldsymbol{\\theta}}\_{u i}, 1\\right)\|}{\|\\ell\\left(\\hat{y}^{\\boldsymbol{\\theta}}\_{u i}, 0\\right)\|}$
> of $w_1$ and $w_0$ are two terms without
> normalization. After dividing by
> $z\_w = \\frac{\|\\ell\\left(\\hat{y}^{\\boldsymbol{\\theta}}\_{u i}, 0\\right)\|}{\|\\ell\\left(\\hat{y}^{\\boldsymbol{\\theta}}\_{u i}, 1\\right)\|} + \\frac{\|\\ell\\left(\\hat{y}^{\\boldsymbol{\\theta}}\_{u i}, 1\\right)\|}{\|\\ell\\left(\\hat{y}^{\\boldsymbol{\\theta}}\_{u i}, 0\\right)\|}$,
> they will be normalized into 0 1.
>
>
> **Q4. In the summary of the main contributions, the authors claim that they are "the first to perform denoising from both positive and negative perspectives in recommendation". According to my understanding, the paper only proposes a method to denoise the negative sampling (avoid false-negative) from the unlabeled data.**
>
> **Response:** Thanks for your insightful question. We are sorry for the ambiguous description of the contribution. Here we indeed tend to illustrate that we are the first to sample both positive and negative data. Specifically, we sample and annotate the unlabeled data based on their loss. Experimental results show that each unlabeled data is successfully annotated by either positive or negative label.

---

### Author Response · Authors · 2022-11-16
**Response to all reviewers**

We sincerely thank all the reviewers for their valuable feedback and overall positive comments. In the original version, there are a few missing details on the method evaluation, as pointed out by the reviewers. Meanwhile, the discussion on the proposed method could be more comprehensive. Therefore, we make the following clarifications to the paper with the feedback from all reviewers.

(1) **We further discuss the proposed method.** Specifically, we clarify the innovation of our method against existing work and explain more about the weights of IDL components. Besides, We clarify the misclassified hard instance that our method can detect and the generalization of our method on other loss functions and backbone.



(2) **We provide more evaluation of the method.** Specifically, we perform a more thorough evaluation of the proposed IG component of the training data partition and study how the IDL component is sensitive to the sampled data. We also explained the baseline and dataset selection.

---

### Decision · Program_Chairs · 2023-01-20

**Decision:**

Reject

**Justification For Why Not Higher Score:**

Many claims made by the authors are not supported by experiments.

**Justification For Why Not Lower Score:**

N/A

**Metareview: Summary, Strengths And Weaknesses:**

In this paper, the authors proposed a meta-learning framework based on a new inverse dual loss to generate gradients to improve the negative sampling quality for recommender systems. Though the high-level idea is interesting, the experimental results fail to support the claims made by the authors. Moreover, some experimental setups are not convincing.

In summary, this paper is not ready for publication in ICLR.